# Expression of PD-1 and PD-L1 in Endometrial Cancer: Molecular and Clinical Significance

**DOI:** 10.3390/ijms242015233

**Published:** 2023-10-16

**Authors:** Mohd Nazzary Mamat @ Yusof, Kah Teik Chew, Nirmala Chandralega Kampan, Mohamad Nasir Shafiee

**Affiliations:** Gynaecologic-Oncology Unit, Department of Obstetrics and Gynaecology, Hospital Canselor Tuanku Muhriz, Faculty of Medicine, Universiti Kebangsaan Malaysia, Kuala Lumpur 56000, Malaysia

**Keywords:** endometrial cancer, PD-1, PD-L1, molecular classification, immune-checkpoint inhibitor, novel targeted therapy

## Abstract

The landscape of diagnosing and treating endometrial cancer is undergoing a profound transformation due to the integration of molecular analysis and innovative therapeutic approaches. For several decades, the cornerstone treatments for endometrial cancer have included surgical resection, cytotoxic chemotherapy, hormonal therapy, and radiation therapy. However, in recent years, the concept of personalised medicine has gained momentum, reshaping the way clinicians approach cancer treatment. Tailoring treatments based on specific biomarkers has evolved into a standard practice in both initial and recurrent therapy protocols. This review aims to provide an in-depth exploration of the current state of molecular analysis and treatment strategies in the context of endometrial cancer, focusing on the immunological aspect of the PD-1/PD-L1 axis. Furthermore, it seeks to shed light on emerging and innovative approaches that hold promise for the future modulation of endometrial cancer treatments. In essence, as researchers delve into the complex molecular landscape of endometrial cancer and harness the understanding of the PD-1/PD-L1 axis, we are paving the way for more targeted, effective, and personalised therapies that have the potential to significantly improve the outcomes and quality of life for patients with this challenging disease.

## 1. Introduction

Endometrial cancer (EC) is the sixth most common malignancy among women worldwide [1]. Geographically, EC is more common in the Northern and Southern Hemispheres, Europe, and Oceania/Australasia, while it is less common in Africa and South and Central Asia. The most common causes of EC include a family history of the disease, menstrual abnormalities, infertility, exposure to oestrogen, the use of hormonal drugs, obesity, diabetes, and a high body mass index [2]. Pathological data, including histology, stage, grade, myometrial invasion, lymph vascular space invasion, and cervical stromal invasion, have been used for decades to determine the recurrence risk of EC. However, expert gynaecologic pathologists may face challenges in classifying the risk based on pathological findings [3,4]. Histology, grade, stage, and risk classification criteria, such as cervical stromal invasion, myometrial invasion, and lymph vascular space invasion varied among reviewers in past therapeutic studies that required central pathology evaluation [5]. As a result, some patients may not receive appropriate care or treatment due to inconsistent risk categorization.

Recent advancements in genome analysis technology have revealed genomic anomalies within EC. Additionally, integrated genomic analyses have identified molecular subgroups that align with prognosis. The most notable approach of integrating molecular characteristics with EC classification by the Cancer Genome Atlas (TCGA) has resolved the numerous limitations in risk stratification. Later, a new model named ProMisE (Proactive Molecular Risk Classifier for Endometrial Cancer) was introduced to improve the limitation of the TCGA methodologies used for immediate clinical application. Several studies showed that this paradigm is effective for diagnostic specimens like endometrial biopsies, curettages, and final hysterectomy specimens. This model was implemented in the European Society of Gynecological Oncology (ESGO), the European Society for Radiotherapy and Oncology (ESTRO), and the European Society of Pathology (ESP) 2020 Guidelines for the management of EC patients based on tumour aggressiveness and recurrence [6,7,8]. These findings are expected to play a significant role in guiding treatment and management decisions for EC [7,8]. Furthermore, diverse biological abnormal changes in pathways have been discerned in EC cells. This has prompted the active development of novel therapeutic drugs and biomarkers, including immunomodulation inhibitors targeting programmed cell death protein 1 (PD-1) or programmed cell death ligand 1 (PD-L1), to address these anomalies. This review aims to comprehensively outline the genomic and molecular alterations observed in EC, while also exploring the potential utility of immunotherapy inhibitors based on these molecular characteristics.

## 2. Materials and Methods

An extensive literature search was conducted by authors to identify relevant studies on various databases (Pubmed, Scopus, and Web of Science). Articles were screened using the following keywords: “endometrial cancer”, “endometrial carcinoma”, “molecular”, “genomic”, “targeted therapy”, “precision medicine”, “programmed cell death 1”, “programmed death ligand 1”, “immunotherapy”, and “immune checkpoint inhibitor”. All articles in English that met review’s objective, as indicated using these keywords, were included without any restriction on the publication year.

## 3. Molecular Classification of Endometrial Cancer

In 2013, a collaborative effort involving multiple agencies and led by the Genome Institute undertook a project to analyse bioinformatics data from TCGA database. This project focused on genomic, transcriptomic, and proteomic datasets obtained through various techniques, including DNA sequencing, a combination of DNA methylation reverse phase protein array, and microsatellite instability analysis. The study encompassed a total of 373 EC cases, comprising 306 endometrioid carcinomas and 66 cases of serous/mixed carcinoma [9]. Notably, the TCGA initiative represents the most extensive genetic research undertaken for EC to date. EC is classified into four distinct genomic categories that are determined by analysing somatic gene mutations, microsatellite instability, and somatic copy number changes within the tumour samples [10].

The first subgroup is the “super mutation”, which displays a significant elevated mutation rate and features a mutant polymerase-ε (POLE) gene with alterations in the polymerase and exonuclease domains [10]. It accounts for about 7–12% of all EC [11,12]. This subtype displays an exceptionally high mutation rate of 232 × 10^−6^ mutations per megabase (Mb) [13]. In a recent study by Jiang et al., distinct immunology-based biomarker signatures were unveiled for various EC subtypes [14]. This investigation revealed that the subgroup characterised by a mutated POLE gene exhibited the lowest innate anti-PD1 resistance expression profile, alongside heightened T-effector and interferon-gamma expression traits. Clinically, this subtype demonstrates a more favourable prognosis [15].

The second subgroup, the “high mutation”, is characterised by microsatellite instability (MSI) resulting from MutL homolog 1 (MLH1) promoter methylation, elevated mutation rates, and stochastic copy number alterations [10,16]. This subgroup is also characterised by a high mutation rate of 18 × 10^−6^ mutations per Mb. The presence of a deficiency in the DNA mismatch repair (MMR) system is a common trait in this group, often attributed to mutations in genes like MLH1, MutS homolog 2 (MSH2), MutS homolog 6 (MSH6), or PMS1 Homolog 2 (PMS2). This MMR deficiency gives rise to the MSI EC phenotype. While somatic mutations of these MMR-related genes, particularly MLH1 promoter methylation, are frequently observed in sporadic MSI EC cases, germ-line mutations of these genes are more commonly found in hereditary EC instances, such as Lynch syndrome patients [17]. Intriguingly, Bellone et al. discovered that among sporadic “high mutation” (MSI-H) EC patients treated with pembrolizumab, those with sporadic MSI-H status exhibited a lower infiltration of CD68+ macrophages within tumour masses and stromal regions compared to Lynch-like MSI-H patients. Additionally, the sporadic MSI-H group demonstrated lower objective response rates (ORR), progression-free survival (PFS), and overall survival (OS) [16].

The third subgroup is the “copy-number low” (low-CN) group, and encompasses the majority of microsatellite stable (MSS) grade 1 and 2 endometrioid carcinomas characterised by low mutation rates (2.9 × 10^−6^ mutations/Mb) [10]. This group exhibits a “no specific molecular profile (NSMP)” and occupies a prognosis position between the POLE-mutation and the high-CN group [18]. It constitutes nearly 60% of low-grade EC cases and represents only 8.7% of high-grade ECs. Notably, mutant phosphatase and tensin homolog (PTEN) and phosphatidylinositol-4,5-bisphosphate 3-kinase catalytic subunit alpha (PIK3CA) genes are identified in 77% and 53% of cases belonging to this EC subgroup, respectively [19]. It manifests a range of clinical symptoms, spanning from mild to aggressive presentations [18].

The fourth subgroup, the “copy-number high” (high-CN) group is characterised by extensive copy number aberrations, relatively low mutation rates, and frequent p53 mutations [10]. Compared to all four EC groups stratified by TCGA, the high-CN group exhibits a poorer prognosis [20]. In the context of the TCGA trial, all patients with serous carcinoma were categorised into the high-CN EC group, and a quarter (25%) of these patients had grade 3 endometrioid carcinoma [21]. Both low-CN and high-CN ECs are traditionally categorised as hypomutated EC, demonstrating low levels of expression for immune-related biomarkers. The molecular classifications of EC based on TCGA are summarised in Table 1.

The Proactive Molecular Risk Classifier for EC (ProMisE) is a novel, validated algorithm-based analytical for EC screening [22]. This algorithm-based protocol employs proteomic immunohistochemistry (IHC) datasets for p53/MMR analysis and POLE mutation assessment to streamline POLE analysis expenses and patient accessibility [22]. The ProMisE workflow commences by determining the presence or absence of two MMR proteins, MSH6 and PMS2, using IHC techniques. This detection helps categorise the patients with EC into the subgroup with deficient mismatch repair (dMMR) if both proteins are absent. Subsequently, for cases with expressed MSH6 and PMS2 proteins, a sequencing evaluation targets the detection of the POLE exonuclease domain mutation (POLE EDM). The detection of this mutation places the patient in the POLE ultra-mutated category. Finally, for patients not falling under dMMR or POLE EDM categories, the patient’s p53 status is assessed using immunohistochemistry IHC to determine whether they exhibit wild-type or null/missense mutations [22,23,24].

Currently, it is recommended to conduct molecular analysis on all cases of EC in accordance with the outlined algorithm. The determination of whether to proceed with molecular testing is contingent on the resources and organizational structure of each centre’s multidisciplinary team. The overarching objective has consistently been to devise a feasible and cost-effective molecular classification that could also be applied to endometrial biopsies or curettages. The biological and molecular insights derived from the tumour play a pivotal role in formulating appropriate therapeutic strategies, determining the extent of surgical intervention, and considering potential adjuvant or molecular therapies.

## 4. Immune Micro-Environment in Endometrial Cancer

The immune system plays a multifaceted role in the normal endometrium, particularly within endometrial epithelial cells. Acting as a part of the mucosal immune system, it serves as a physical barrier and is involved in the production of defensins, other immune mediators, and antigens [25,26,27,28]. In a normal endometrium, both the innate and adaptive immune systems are pivotal in eliminating pathogens, generating inflammatory cytokines, and regulating immune responses [25,29,30]. These functions are intricately regulated by sex hormones, particularly oestradiol and progesterone, which exhibit variations corresponding to the fluctuations in the menstrual cycle [25,31,32]. The immune system of the endometrium comprises various components, all with the dual purpose of maintaining normal physiological processes. It creates an environment that suppresses immune responses to prevent the mother’s immune system from rejecting the foetus, while simultaneously protecting the compromised endometrium from potential infections during menstruation [25,33]. Under normal circumstances, PD-1 serves to inhibit autoimmunity, restrict damage from infections to healthy tissues, and promote self-tolerance [34,35]. Elevated PD-1 expression on T cells can impact their ability to combat cancer and infectious diseases [34,36,37].

A cancerous endometrium exhibits a distinctive microenvironment where carcinogenic substances can directly impact immune-related signalling pathways or the host’s defensive inflammation, leading to changes in immunological balance through tumour-induced immunoediting. Throughout the process of carcinogenesis, the endometrial immune response can both promote and inhibit tumour growth. The “cancer immunosurveillance” hypothesis, initially proposed by Burnet and Thomas in the late 1950s, posited that tumours in immunocompetent hosts trigger an immune system response aimed at restricting malignant cell proliferation [38,39]. This conceptual framework aimed to connect the immune system with tumour progression. However, subsequent experimental findings contradicted this hypothesis, leading to ongoing debates. Consequently, the cancer immunosurveillance concept fell out of favour [40]. In 2002, Ikeda, Old, and Schreiber introduced a more intricate model known as “Cancer Immunoediting”, suggesting that the immune system can simultaneously impede and accelerate tumour growth [38,41]. This model comprises three phases: elimination, equilibrium, and escape. This traditional model will be employed to illustrate the reciprocal interactions between EC tumours and immunity. During the elimination phase, both innate and adaptive immune responses identify and eliminate EC cells through cytotoxic mechanisms [42]. Dendritic cells (DCs) play a role by phagocytosing and processing “altered self” EC cells and “non-self” antigens under conditions of stress and harm [43,44]. DCs then stimulate and present tumour-associated antigens to generate CD8+ cytotoxic T lymphocytes (CTLs) and CD4+ T cells [44,45]. CD8+ CTLs directly destroy EC cells, while CD4+ T helper cells activate specific B cell responses for both humoral and cytotoxic immune reactions [46]. The cancer immunoediting process concludes when the immune system successfully eliminates all EC cells [47]. Despite this elimination phase, a small fraction of malignant cells may survive and enter a state of equilibrium. During the “equilibrium” phase, EC cells and the immune system establish a dynamic balance, resulting in a period of temporal biological equilibrium [48]. Numerous dormant EC cells can remain latent in patients for extended periods [49]. This period is characterised by complex interactions between the immune system and EC, ultimately determining the fate of the tumour. If EC cells manage to create an immunosuppressive microenvironment, they transition into the “escape” phase, evading immune control [50]. As a result, EC cells regain their ability to proliferate and form distant metastases [47].

Understanding the mechanism of the shift from the equilibrium to the escape phase holds the potential to inform the development of effective immunotherapies. Recent research suggests that tumour cells release factors such as vascular endothelial growth factor (VEGF), transforming growth factor beta (TGF-β), and indoleamine 2,3-dioxygenase (IDO) to suppress immune cells [51,52,53]. Tumour cells can evade immunosurveillance by shedding tumour antigen and major histocompatibility complex class I molecules or by employing immune-inhibitory mechanisms involving regulatory T cells (Tregs) and myeloid-derived suppressor cells [54]. Immune checkpoint pathways play a crucial role in inhibiting activated T-cells through negative regulatory pathways, and these pathways must be upregulated by tumour cells to evade immune surveillance [55]. In EC, the immune system requires two signals to activate naïve T-cells and prompt them to recognise and target tumour cells [56]. The initial signal involves T-cell receptors (TCR) binding to antigenic peptide-bearing major histocompatibility complexes on EC cells, though this signal alone is insufficient to activate T-cells. The subsequent signal is generated when costimulatory molecules such as CD80 and CD86 (also known as B7-1 and B7-2) on the antigen-presenting cell (APC) interact with T-cell ligands like CD28 [47,57,58]. Immune checkpoint pathways play a role in negatively regulating this two-signal activation process, enabling EC cells to evade immune attack [59,60,61]. Key components of immune checkpoint signalling, such as PD-1, CTLA-4, and PD-L1, are expressed by T cells and other immune cells on the surface of EC cells. CTLA-4 competes with CD28 for B7 ligands, thereby dampening the costimulatory signalling of the CD28/B7 axis (the second signal) [61,62,63]. PD-1/PD-L1 engagement triggers the recruitment of tyrosine phosphatase SHP2, which dephosphorylates proximal signalling regions of T-cell receptors [64]. This dephosphorylation results in a negative costimulatory effect, suppressing T-cell activation [61]. These immune checkpoint signals have emerged as crucial targets for novel immunotherapies.

## 5. Expression of PD-1 and PD-L1 in Endometrial Cancer

PD-1 is a cell surface molecule consisting of 288 amino acids. It is classified as a membrane protein within the immunoglobulin superfamily in humans. PD-1 functions to suppress both adaptive and innate immune responses. This protein is present in various immune cell types, including activated T cells, natural killer (NK) cells, B lymphocytes, macrophages, DCs, and monocytes [65,66]. Notably, tumour-specific T lymphocytes exhibit a high expression of PD-1 [67]. Transcription factors, specifically nuclear factor of activated T cells (NFAT), forkhead box protein (FOX) O1, and interferon (IFN) regulatory factor 9 (IRF9), hold the capacity to initiate the transcription process of PD-1 [66,68]. Critical in the regulation of the PD-1 gene’s expression are the upstream regulatory regions B and C, designated as CR-B and COR-C. The CR-C region harbours a binding site associated with NFATc1 (NFAT2) in TCD4 and TCD8 cells [66,69]. In contrast, the protein c-FOS engages with specific sites within the CR-B region. This interaction amplifies the production of PD-1 when T-cell receptors are activated upon antigen recognition in naïve T cells. PD-1 exhibits a dualistic nature, encompassing both beneficial and detrimental effects. On the advantageous side, it plays a pivotal role in curbing ineffective or harmful immune responses and upholding immunological tolerance. However, PD-1 activation hampers the protective immune response, contributing to the advancement of malignant cells [70]. PD-1 is a member of the CD28 family and is encoded by the PDCD1 gene situated on chromosome 2q37.3. This protein has two distinct ligands: PD-L1 and programmed cell death ligand 1 (PD-L2). These ligands exhibit diverse expression patterns and are, respectively, encoded by the CD274 gene and the PDCD1LG2 gene, both located on chromosome 9p24.1 [71].

PD-L1 is typically detected in macrophages, activated T cells, B cells, DCs, and certain epithelial cells, particularly in the presence of inflammatory stimuli [66,72]. Additionally, tumour cells exploit PD-L1 expression as a means to evade anti-tumour responses, representing an adaptive immune mechanism [73]. The presence of PD-L1 is associated with an immunological microenvironment characterised by an abundance of CD8 T cells, the secretion of Th1 cytokines and chemical factors, and the generation of interferons and distinct gene expression patterns [74]. Prior studies have provided evidence that interferon-gamma (IFN-γ) triggers an elevation in PD-L1 expression in ovarian cancer cells, contributing to disease progression. Conversely, the inhibition of IFN-γ receptor 1 has been shown to reduce PD-L1 levels in acute myeloid leukaemia mouse models. This reduction is achieved through the mitogen-activated protein kinase (MEK)/extracellular signal-regulated kinase (ERK) and myeloid differentiation primary response 88 (MYD88)/Tumor necrosis factor receptor-associated factor 6 (TRAF6) pathways [11]. The induction of protein kinase D isoform 2 (PKD2) by IFN-γ plays a crucial role in PD-L1 regulation. Inhibiting PKD2 activity suppresses PD-L1 expression, thereby enhancing immune response effectiveness against tumours. Interferon-gamma (IFN-γ) is produced by NK cells through the activation of the Janus kinase (JAK)1, JAK2, and signal transducer and the activator of the transcription (STAT)1 pathway. This leads to an upregulation of PD-L1 expression on the tumour cell surface [75]. Studies conducted on melanoma cells have demonstrated that T cell-secreted IFN-γ, via the JAK1/JAK2-STAT1/STAT2/STAT3-IRF1 pathway, can modulate PD-L1 expression. Both T cells and NK cells release IFN-γ, leading to the induction of PD-L1 expression on target cell surfaces, including tumour cells [76].

PD-L1 functions as a promoter of tumour growth within cancer cells by engaging specific receptors and initiating signalling pathways that encourage cell proliferation and survival [77]. This discovery contributes further evidence to the involvement of PD-L1 in the subsequent progression of tumours. Furthermore, it has been demonstrated that PD-L1 exerts non-immune proliferative effects on various types of tumour cells. For instance, the activation of PD-L1 has been observed to trigger epithelial-to-mesenchymal transition (EMT) and the acquisition of stem cell-like characteristics in renal cancer cells. This implies that the intrinsic pathway of PD-L1 plays a role in facilitating the advancement of kidney cancer [78].

PD-L1 is one of several ligands capable of binding to the PD-1 receptor. In a process where tumour cells elevate PD-L1 production, this protein interacts with PD-1 in T cells, initiating a co-inhibitory signal within these T cells [13,79]. This mechanism enables tumour cells to evade elimination through T-cell cytolysis, thereby fostering tumour progression (Figure 1). The route is a primary focal point for immune checkpoint inhibitors (ICIs). The PD-1 or PD-L1 inhibitors inhibit the interaction between the PD-L1 and PD-1 receptor. Hence, this prevents cancer cells from escaping the immune system by reactivating the T-cell-mediated process of eliminating tumour cells (Figure 2). Recent investigations have unveiled that anti-PD-1/PD-L1 first line therapy yields response rates varying between 20% and 65% in PD-L1-positive tumours in various cancers, including EC [24,80,81,82]. Conversely, tumours lacking PD-L1 expression exhibit response rates ranging from 0% to 17% across diverse tumour types [79]. The significance of PD-L1 expression within the tumour microenvironment is recognised as a pivotal biomarker for identifying individuals who are more likely to benefit therapeutically from immunotherapy [13].

The expression levels of PD-1 and PD-L1 in EC are remarkably high, with PD-1 being expressed in approximately 60–65% of EC cases and PD-L1 in a range of 25–70% [19,83,84,85,86,87,88]. These levels are among the highest reported among gynaecological malignancies. However, ongoing debates persist around the expression patterns of PD-L1 across different molecular subtypes of EC. For instance, in a study by Howitt et al., it was discovered that the frequency of PD-L1 expression was higher in POLE and MSI tumours compared to MSS tumours, particularly in intraepithelial immune cells [89]. This comparison was made between cases exhibiting the presence versus complete absence of PD-L1 expression.

A study conducted on a cohort of 132 patients diagnosed with MSS, grade 2 endometrioid carcinoma highlighted a specific subgroup within MSS endometrioid carcinomas exhibiting elevated PD-L1 expression [19]. The results from this study indicated that within the cohort of MSS tumours, approximately 48% of them exhibited positive PD-L1 expression. Moreover, within this group, about 16% expressed a more diffuse or notably intense PD-L1 expression. This subgroup of MSS EC with elevated PD-L1 expression shared an interesting similarity with a distinct class of EC known as MSI EC. These cancers are characterised by a particular microsatellite genetic instability. Remarkably, the subgroup identified in the study not only showed elevated PD-L1 expression, but also demonstrated elevated levels of tumour-associated CD3+ and CD8+ lymphocytes. This heightened immune cell presence is a shared feature with MSI EC. In 2019, the European Society for Medical Oncology (ESMO) conducted a comprehensive analysis examining the correlation between MSI status and PD-L1 expression across various cancer types, including EC [90]. This analysis provided valuable insights into how these two factors interplay in various cancers and potentially influence treatment strategies. The discoveries from both the initial study on the MSS grade 2 endometrioid carcinoma subgroup and the subsequent broader analysis by ESMO shed light on the intricate relationship between PD-L1 expression, MSI, and immune cell infiltration within different types of cancers.

In EC, only a small fraction, comprising 3.1% of patients, were found to exhibit MSI and a positive PD-L1 status. Among the diverse malignancies assessed, the group of patients with MSI-H combined with a positive PD-L1 status constituted a relatively small proportion. A study by Vanderwalde et al., involving 11,348 cases across 23 different cancer types, found that the prevalence of PD-L1-positive cases was 25.4% in the entire population [91]. However, within the group of patients with MSI-H, only 26% exhibited PD-L1 positivity. The elevated expression of PD-L1 has been linked to a higher prevalence of patients with high-grade and non-endometrioid EC, as reported in previous studies [17,92,93]. Notably, an association has been established between higher levels of PD-L1 expression in tumour cells and immune cells and the presence of dMMR [92,93,94]. Furthermore, TCGA classification has provided new insights into molecular subtype mutations that altered EC’s immunological profile. Previous studies have demonstrated that PD-L1 expression is significantly more pronounced in the POLE and dMMR subgroups compared to the NSMP and p53 mutation subgroups [17,93,95]. Several studies, including those by Li et al. and Zong et al., have reported associations between PD-L1 expression in immune cells and factors such as deep myometrial invasion, the presence of lymphovascular space invasion (LVSI), and the histological subtype of non-endometrioid EC. However, these associations were not consistently observed in tumour cells [92,93]. Additionally, a higher expression of PD-L1 in tumour cells was more frequently observed in high-grade tumours compared to low-grade tumours [92,93,96]. These findings highlighted the complex relationships between microsatellite instability, PD-L1 expression, and various clinicopathological features in EC. Understanding these associations provides valuable insights into the complex molecular landscape that governs the behaviour of different subtypes of EC, as well as prognostic indicators and potential treatment strategies.

## 6. Clinical Significance: Prognostic Role and Immune Checkpoint Inhibitors

The relationship between PD-L1 expression and its impact on the prognosis of endometrial cancer patients has been extensively explored, yielding inconsistent findings when correlated with tumour cells or immune cells and their associations with survival [83,84,87,93,97]. Recent studies have contributed to the complexity of this matter. A comprehensive study involving a large cohort of 833 patients conducted by Zong et al. revealed that PD-L1 expression in TCs was associated with a favourable prognosis in patients with FIGO stages II–IV and non-endometrioid endometrial cancer, but this association was not observed in immune cells [93]. In contrast, another study by Zhang et al. demonstrated that high PD-L1 expression in TCs acted as an independent predictor of favourable OS, while elevated PD-L1 expression in ICs correlated with worse OS outcomes [84]. Earlier research by Yamashita et al. similarly indicated that PD-L1 expression in TCs was linked to prolonged progression-free survival [98]. On the contrary, Chew et al. reported contradictory results, showing that PD-L1 expression in tumour cells was significantly associated with poor survival [87]. Additionally, Kucukgoz Gulec et al. found that PD-L1 expression in non-endometrioid endometrial cancer was associated with shorter survival in tumour cells [97]. A key insight from Zong et al.’s work is that high-risk features in endometrial cancer patients, particularly those with non-endometrioid endometrial cancer, increase the likelihood of distant metastasis and disease recurrence. Consequently, studies in cancer immunology and immunotherapy should prioritise these subgroups. Their findings highlighted the distinct prognostic roles of PD-L1 expression in tumour cells and immune cells. This underscores the importance of separately assessing these expressions, rather than using a combined positive score (CPS) technique, to predict patient outcomes and determine eligibility for immunotherapy [93].

The primary treatment for high-risk EC patients, as per the FIGO staging classification, involves surgery coupled with adjuvant chemotherapy or radiotherapy [99]. Immunotherapy, which inhibits immunological checkpoints to target various biological pathways, could potentially benefit EC patients. In around 30% of primary EC patients with MSI-H or dMMR tumours, PD-1 and its ligands are specifically targeted [100]. However, the clinical application of immunotherapy is constrained, and further research is required to evaluate its efficacy. Despite advancements in therapy, the survival and quality of life for EC patients remain unchanged. The molecular heterogeneity underlying tumour invasion and metastasis might elucidate why patients with similar clinicopathological characteristics experience differing disease outcomes. Consequently, clinicians necessitate enhanced disease classification tools to refine diagnosis, treatment, and prognosis [101].

Nivolumab and pembrolizumab (both are PD-1 inhibitors) emerged as the earliest immune checkpoint inhibitors to gain Food and Drug Administration (FDA) approval in 2014, catering to patients with advanced melanoma who displayed an inadequate response to first-line treatments [102,103]. In 2015, these inhibitors received clearance for use as first-line therapy, yielding response rates of 26–40% among patients with advanced melanoma. Impressively, these treatments demonstrated superior survival rates compared to traditional chemotherapy options [104,105,106]. Subsequent investigations unveiled that PD-1 immune checkpoint inhibitors exhibited heightened effectiveness, a more favourable safety profile, and sustained response over an extended period [106,107,108]. Additionally, focusing on the MMR status, pembrolizumab emerged as the pioneering immune checkpoint inhibitor capable of enhancing the overall response rate (ORR) in cases of metastatic or recurrent colorectal and non-colorectal malignancies characterised by dMMR [109,110].

Just like in other cancer types, the exploration of immunomodulation in EC has been an active area of research. In 2017, the FDA granted approval for pembrolizumab to be used in cases of unresectable or metastatic solid tumours characterised by MSI-H or dMMR. various immune checkpoint inhibitors have been investigated, including anti-PD-1 and anti-PD-L1 agents such as nivolumab, durvalumab, and dostarlimab. The ORRs for these treatments have been documented at 25%, 27%, 43%, and 42%, respectively [111,112,113,114]. This highlighted the potential of immunotherapy to elicit meaningful responses in a subset of EC patients, offering a promising avenue for those with advanced or recurrent disease.

The demonstrated efficacy and acceptable safety profile of monotherapy immune checkpoint inhibitors represents significant achievements. However, there remains room for improvement in terms of achieving higher response rates. This has led to the exploration of combined therapy approaches in clinical trials, aimed at identifying potential modifications that could enhance response rates. For instance, the application of immune checkpoint inhibitors has so far been limited to advanced or recurrent endometrial cancer cases that have undergone at least one line of platinum-based chemotherapy. The question of whether immunotherapy could enhance outcomes for individuals with dMMR endometrial cancer in conjunction with traditional adjuvant therapies remains unanswered. This investigation is being undertaken in the RAINBO clinical trial program, wherein patients with stage II/III endometrial cancer are being randomly assigned to receive radiation therapy alone or radiation therapy combined with durvalumab (PD-L1 inhibitor) [115]. In addition to conventional chemoradiation and chemotherapy approaches, the ADELE trial is evaluating the potential of an immune checkpoint inhibitor named Tislelizumab (PD-1 inhibitor), in combination with chemotherapy as a post-chemoradiation intervention for high-risk endometrial cancer cases [116].

The KEYNOTE-146 study has combined pembrolizumab with lenvatinib, a multikinase inhibitor designed to target vascular endothelial growth factor receptors 1–3, in the context of treatment for advanced solid tumours. Among the cohort of patients with EC, the trial exhibited a 64% ORR in the dMMR group and a 36% ORR in the MMR-proficient group [117]. Subsequently, in 2019, the FDA granted approval for the use of pembrolizumab in combination with lenvatinib for patients with MMR-proficient EC who experienced disease progression after prior systemic treatment. A significant breakthrough has been made through the KEYNOTE-775 phase-III trial, which undertook a comparison between pembrolizumab and lenvatinib against doxorubicin or paclitaxel in advanced or recurrent EC patients with previous platinum-based treatment. Notably, the results presented at the 2021 Society of Gynecologic Oncology (SGO) annual meeting revealed substantial enhancements in progression-free survival (7.2 months vs. 3.8 months) and overall survival (18.3 months vs. 11.4 months) within the pembrolizumab plus lenvatinib arm, as opposed to the chemotherapy group, encompassing both MMR-proficient and dMMR EC cases [118].

At the ASCO 2020 conference, a novel treatment approach involving nivolumab (PD-1 inhibitor) and cabozantinib (tyrosine kinase inhibitor) was presented for the management of recurrent EC in patients who had previously undergone systemic therapy. The patient group, who had received substantial prior treatments, exhibited a 25% ORR and a considerable 69% clinic benefit rate when treated with this combination. An intriguing finding emerged from a subgroup analysis involving 21 patients who had experienced disease progression following treatment solely with ICIs. When these patients were re-challenged with the combination of nivolumab and cabozantinib, the results indicated an ORR of 5 out of 21 and stable disease observed in 12 out of 21 individuals. This observation suggests that re-challenging the ICIs alongside tyrosine kinase inhibitor could potentially overcome resistance mechanisms. Furthermore, the combination therapy demonstrated notable activity in the context of recurrent carcinoma-coma, a particularly aggressive subtype of EC. Among a group of nine patients with this subtype, an ORR of one out of nine was achieved, and stable disease was observed in four out of nine cases [119].

For advanced or recurrent EC, three clinical trials are currently evaluating the efficacy of ICIs in combination with first-line carboplatin and paclitaxel chemotherapy. These trials include AtTEnd/ENGOT-en7, which utilises atezolizumab (PD-L1 inhibitor); RUBY trial, which employs dostarlimab (PD-1 inhibitor); and NRG-GY018, using pembrolizumab (PD-1 inhibitor). The results from both the RUBY trial and the NRG-GY018 trial have demonstrated that the combination of immune checkpoint inhibitors with standard chemotherapy significantly increases progression-free survival in patients with primary advanced or recurrent endometrial cancer [120,121]. Notably, there is a substantial benefit observed in patients with deficient mismatch repair (dMMR) status, indicating the potential of this approach in improving outcomes for this specific subgroup of EC patients.

## 7. Discussion

In recent years, there have been substantial advancements in the research surrounding PD-1/PD-L1 expression in endometrial cancer, shedding light on its complex role within the tumour microenvironment. However, despite these advancements, there remain certain aspects that have proven elusive, indicating gaps in our understanding of this intricate phenomenon. One notable challenge in the field of PD-1/PD-L1 expression in EC is the presence of inconsistent and inconclusive evidence. While some studies have reported significant levels of expression within EC, others have found relatively lower expression levels [84,85,88,122]. This disparity could be attributed to various factors, such as variations in the patient population, tumour heterogeneity, and differences in laboratory techniques used to assess PD-1/PD-L1 expression.

One particularly interesting aspect that continues to puzzle researchers is the lack of PD-1/PD-L1 expression observed in certain subgroups of EC. Despite the overarching understanding that PD-1/PD-L1 expression serves as a critical mechanism for immune evasion by tumour cells, some subsets of endometrial cancer patients seem to exhibit limited or even absent PD-1/PD-L1 expression. This raises questions about the underlying mechanisms that contribute to this phenomenon. Researchers are actively investigating whether there are specific genetic mutations, epigenetic alterations, or microenvironmental factors that might account for the absence of PD-1/PD-L1 expression in these cases. Moreover, the complex interaction between PD-1/PD-L1 expression and other components of the immune microenvironment further complicates the picture. Immune cells, cytokines, and other immune-modulating factors all contribute to the regulation of PD-1/PD-L1 expression on both tumour cells and immune cells [36]. Understanding these complex interactions and how they influence PD-1/PD-L1 expression patterns is crucial for developing targeted therapies that effectively disrupt immune evasion mechanisms.

The lack of a comprehensive understanding of PD-1/PD-L1 expression in certain subgroups of EC points to the need for more comprehensive and interdisciplinary research approaches. Collaborations between clinicians, immunologists, geneticists, and other specialists are essential to unravel the underlying complexities of PD-1/PD-L1 expression. Advanced techniques in genomics, single-cell analysis, and immune-profiling offer promising avenues to explore the mechanisms governing PD-1 expression variation across different endometrial cancer subtypes. The challenge of comprehending the heterogeneity of PD-1/PD-L1 1 expression within different patient populations and tumour subtypes underscores the intricate nature of immune responses in cancer [83]. Continued research efforts aim at deciphering the molecular and cellular underpinnings of PD-1/PD-L1 expression variations will undoubtedly contribute to the development of more effective and personalised therapeutic strategies for endometrial cancer patients.

PD-1/PD-L1 inhibitor therapy stands as the cornerstone of cancer immunotherapy, revolutionizing the approach to cancer treatment. The fundamental principles and clinical insights derived from this therapy have fundamentally transformed how clinicians view and treat cancer through immunotherapy. The knowledge gained from the success of ICIs has far-reaching implications, not only for the development of novel treatments but also for the understanding of the complex interplay between the immune system and cancer progression. There are three fundamental principles and key clinical findings that have emerged from the use of ICIs like anti-PD-1/PD-L1: tumour site immune modulation, targeting tumour-induced immune defects, and repairing ongoing tumour immunity [123]. ICIs have demonstrated the capacity to modulate the immune response at the site of the tumour [124]. By blocking the PD-1/PD-L1 pathway, these therapies unleash the human body’s immune cells to mount a more effective elimination against cancer cells within the tumour microenvironment. Cancer often creates an immunosuppressive environment that hinders the body’s natural immune response. ICIs are designed to target and overcome these tumour-induced immune defects, allowing the immune system to recognise and destroy cancer cells effectively. ICIs not only enhance the immune response, but also have the potential to repair and strengthen the body’s ongoing antitumor immunity. This means that these medications can be effective in various stages of cancer, including advanced and metastatic disease of EC.

The mechanisms leading to the most suitable therapeutic options of ICIs in EC remain uncertain. Several studies have suggested that high levels of PD-1/PD-L1 expression are indicative of a favourable prognosis [83,84,93,125]. However, it is crucial to recognise that tumours can develop resistance to therapeutic interventions by dynamically increasing the expression of PD-1/PD-L1. The relationship between the level of PD-1/PD-L1 expression and the therapeutic impact is not straightforward, emphasizing the need for appropriate treatment strategies. Assessing PD-1/PD-L1 expression is widely recognised as a critical step in implementing PD-1/PD-L1 inhibitors. Initially, it is essential to determine the presence of PD-1/PD-L1 expression to decide whether ICIs are appropriate for the tumours. Equally important is the continuous monitoring of fluctuating levels of PD-L1 expression during treatment [126,127].

Combining PD-1/PD-L1 inhibitors with other therapeutic agents or chemotherapy represents a promising therapeutic approach. However, it is important to note that the predictive value of increasing blockade on the effectiveness of a particular phenomenon is not guaranteed, as there exists a threshold beyond which opposite effects may be observed. The human immune system operates within a delicate equilibrium involving multiple molecules, immune cells, and effectors [127]. Consequently, the role of any one pathway cannot be considered in isolation, highlighting the complexity of immunotherapy and the need for further research to unravel its complexities.

Indeed, in this era of rapidly advancing medical science, one thing is abundantly clear: the enigma of ICIs in EC is far from being completely unravelled. The complexity of the immune system’s interactions with tumours and PD-1/PD-L1 inhibitors continue to present a great challenge to researchers. Hence, the need for more clinical trials is paramount. These trials serve as the crucibles in which scientific hypotheses are tested, and the true potential of ICIs in EC can be explored. They provide the structured environment necessary to collect robust data, analyse outcomes, and draw meaningful conclusions. Moreover, clinical trials allow for the evaluation of ICIs across various patient populations, disease stages, and treatment combinations, yielding valuable insights that can guide future therapeutic strategies.

Beyond data summary and findings, the strength of this review includes the studies’ critical appraisal and some level of data synthesis, which highlight the EC molecular classifications based on the comprehensive molecular profiles in predicting treatment efficacy and EC management, rather than just depending on conventional classification. This review has been constructed based on numerous studies, and highlights the importance of molecular classification. It also presents recent research and points out any identified knowledge gaps, particularly in the areas of EC and immunotherapy. Further studies on immunotherapy and associated gene profiles in EC are necessary. An even more robust approach would involve the metadata of complete epigenomics, transcriptomics, and microenvironmental features. Such comprehensive data can pave the way for tailored targeted therapies and the modulation of novel treatments.

## 8. Conclusions

The expression of PD-1/PD-L1 in EC plays a crucial role in modulating immunotherapy. Its integration with molecular classification provides a reliable system for classification, with significant prognostic and therapeutic implications. Priority should be given to clinical trials that focus on molecularly defined EC to evaluate treatment efficacy within biologically similar tumours and improve outcomes in this disease site. In this new era, it is anticipated that there will be enhanced delineation within each molecular subtype, leading to advancements in systemic therapy for EC.

## Figures and Tables

**Figure 1 ijms-24-15233-f001:**
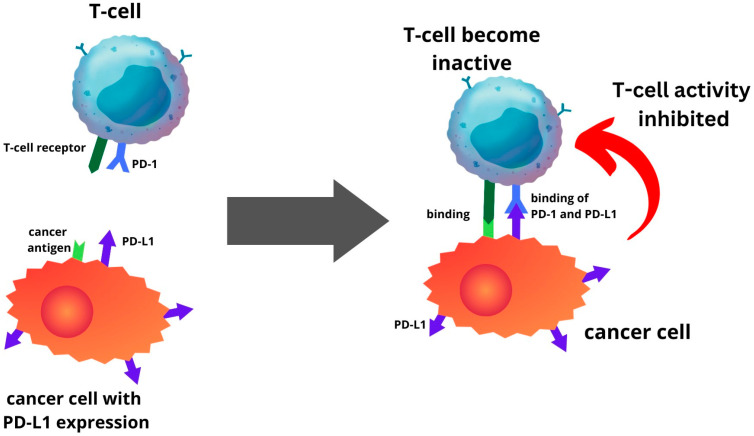
Endometrial cancer cells with PD-L1 expression will be able to inhibit T-cell activity by binding the PD-L1 to PD-1 of T-cell.

**Figure 2 ijms-24-15233-f002:**
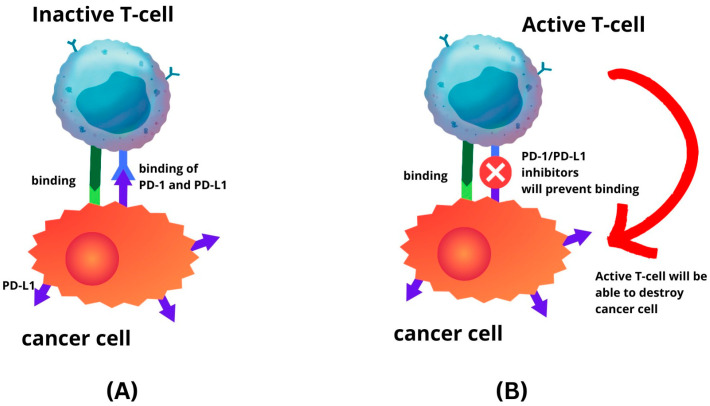
Immune-checkpoint inhibitor mechanism. (**A**) Binding of PD-L1 to PD-1 will inhibit T-cell activity; (**B**) as the result of PD-1 and PD-L1 inhibitors, T-cell will remain active.

**Table 1 ijms-24-15233-t001:** Summary of TCGA molecular classification in endometrial cancer and its associated features.

Molecular Classes	Copy Number High (High-CN)	Copy Number Low (Low-CN)	Microsatellite Instability (MSI)	POLE-Mutant
Associated gene mutation	TP53 PIK3CA FBXW7 PPP2R1A PTEN	PTEN PIK3CA CTNNB1 ARID1A PIK3R1	PTEN PIK3CA PIK3R1 RPL22 ARID1A	POLE DMD CSMD1 FAT4 PTEN
Mutations	Low	Low	High	Extremely high
Somatic copy number alteration	High	Low	Low	Very low
Associated histology	Serous type	Endometroid grade 1–2	Endometroid grade 3	Endometroid grade 3
PD-L1 expression	Low	Low	High	High
Prognosis	Poor	Good/intermediate	Intermediate	Good
Benefit of PD-1/PD-L1 inhibitors	No	No	Yes	Yes

## Data Availability

Not applicable.

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
