# Peer review of "Expression of PD-1 and PD-L1 in Endometrial Cancer: Molecular and Clinical Significance"

_ijms, 2023, doi:10.3390/ijms242015233_

Round 1
Reviewer 1 Report
I read with great interest the manuscript, which falls within the aim of this Journal and offers a high-quality overview of the topic.
This article is very timely since there is a high interest in the field of molecular assessment of endometrial cancer.
Methodology is accurate and conclusions are supported by the data analysis. The figures are clear and interesting.
Although the manuscript can be considered already of high quality, I would suggest taking into account the following minor recommendations:
- I suggest another round of language revision, in order to correct few typos and improve readability.
- I find it interesting to discuss in deep, in the introduction, the new molecular classification of endometrial cancer and its importance for tailoring the approach to this important disease ( see PMID: 36979434).
- The authors could consider adding a summary table.
- The authors have not adequately highlighted the strengths and limitations of their study. I suggest better specifying these points.
Minor editing of the English language is required to make the work clearer and more readable.
Reviewer 2 Report
''Expression of PD-1 and PD-L1 in Endometrial Cancer: Molecular and Clinical Significance''
The manuscript presented for review consists of 16 pages with 127 references. The authors focused on the issue concerning endometrial cancer, its molecular aspects and prepared a summary of the current knowledge. The study fits the Journal scope. The topic is relevant for the field. 2 figures are included. They properly present mechanisms. The legends are described. The manuscript is divided into 7 sections. The study is clear and well-structured. Abstract is presented in 189 words. The aim is defined. A Background is good arranged with sufficient literature review.
- I would suggest to add a chapter which describe material and methods (inclusion/exclusion criteria of the study, how many studied did you analysed, the reason that some researches were not included etc).
References:
Literature is up-to-date. It reffers to the whole context of the study.
-I would recommend to modify the style of references according to IJMS guidelines.
Keywords are adequate and refer to the whole context.
- I would add:
''novel therapy targets''
English:
English language is understandable.
In my opinion this manuscript may be accepted after minor revision.
